# Students from a Public School in the South of Chile with Better Physical Fitness Markers Have Higher Performance in Executive Functions Tests—Cross-Sectional Study

**DOI:** 10.3390/bs13020191

**Published:** 2023-02-20

**Authors:** Jesús Alonso-Cabrera, Franco Salazar, Jorge Benavides-Ulloa, María Antonia Parra-Rizo, Rafael Zapata-Lamana, Caterin Diaz-Vargas, Jaime Vásquez-Gómez, Igor Cigarroa

**Affiliations:** 1Departamento de Matemáticas y Estadística, Universidad del Norte, Barranquilla 081008, Colombia; 2Escuela de Kinesiología, Facultad de Salud, Universidad Santo Tomás, Los Ángeles 4440000, Chile; 3Faculty of Health Sciences, Valencian International University (VIU), 46002 Valencia, Spain; 4Department of Health Psychology, Faculty of Social and Health Sciences, Campus of Elche, Miguel Hernandez University (UMH), 03202 Elche, Spain; 5Escuela de Educación, Universidad de Concepción, Los Ángeles 4440000, Chile; 6Centro de Investigación de Estudios Avanzados del Maule (CIEAM), Universidad Católica del Maule, Talca 3460000, Chile; 7Laboratorio de Rendimiento Humano, Grupo de Estudios en Educación, Actividad Física y Salud (GEEAFyS), Universidad Católica del Maule, Talca 3460000, Chile

**Keywords:** physical aptitude, Chile, cognition, healthy lifestyle, students, executive function

## Abstract

In the past few years, the level of physical fitness in children has decreased globally. According to the SIMCE test carried out in 2015, 45% of 8th year students in Chile were overweight. Moreover, international studies have shown that being overweight is associated with the development of chronic illnesses, negatively affecting cognitive mechanisms and processes. Nevertheless, there is little to no evidence that analyzes the relationship between physical fitness and executive functions in students, at a national level. The aim was to analyze the relationship between cardiorespiratory, musculoskeletal, and motor fitness, and performance in an executive functions test, in students from a public school in the south of Chile. A qualitative, descriptive –correlational, non-experimental, and cross-sectional approach was used. In total, 100 students between 9 and 12 and 11 months of age from a public school in the south of Chile completed the physical fitness assessments through the ALPHA fitness test, and 81 students completed the executive function assessments through the ENFEN test. It was evidenced that students who achieved a longer duration of time and a later stage in the Course Navette test, more centimeters in the standing broad jump (SBJ) test, and a shorter duration in the 4 × 10 shuttle run obtained a better score in the gray trail test. Additionally, students who presented a stronger dominant handgrip scored higher in the colored trail tests. We conclude that students who show a higher level of physical fitness also present a better development of executive functions such as working memory and inhibitory control. In addition, these results suggest physical condition is a factor to consider for better cognitive and school performance.

## 1. Introduction

Students between 9 and 13 years old find themselves in a transition stage of their lives, in which healthy habits can be built and start to develop [1,2]. Healthy habits are attitudes and patterns of behavior that can be fostered by the family and the school environment [3]. Hence, numerous governmental programs have been created in Chile to counteract the effects of sedentariness since it currently represents one of the biggest public health problems, bringing negative and harmful consequences, such as obesity, cardiovascular diseases, and high levels of mortality [4]. In this respect, a study carried out in 2015 by Quebec Adipose and Lifestyle Investigation in Youth (QUALITY), established a relationship between high levels of sedentariness with a decrease in levels of physical activity, and high levels of overweight and obesity in Canadian children between 8 and 10 years of age [5], as well as in Latin America [6]. In the Chilean context, national reports such as the quality of education measurement system (SIMCE 2015) pointed out that around 10,000 eight grade students, in other words 45% of students in this level, were overweight [7]. Consequently, it is suggested that overweight or obese children are, in the long term, at higher risk of illnesses such as hypertension, type 2 diabetes, and hyperlipidemia [8].

Owing to current healthy habits in Chilean society, simpler and easier indicators to measure integral health in school students are necessary. In this context, the measurement of health-related quality of life (HRQOL) is increasingly important as a means of monitoring population health status over time, of detecting subgroups within the general population with poor HRQOL, and of assessing the impact of public health interventions within a given population. While quality of life research in adults has progressed over the past years, health-related quality of life research in children is a recent field. For children and adolescents, it is important to understand the impact of their health conditions on health-related quality of life in order to plan, act upon, and improve prevention and care [9,10]. Thus, two parameters of HRQOL begin to be important in the school setting: physical fitness and executive functions.

Physical fitness has been recommended as a good indicator since not only does it reflect general health in students, but it is also a measure to describe the status of the main organic functions that intervene in body movement [11]. Through time, the concept of physical fitness has evolved from being mainly related to strength and motor abilities to being closely related to health [12]. It is paramount to measure physical fitness regarding health since an inverse relationship between physical health status and morbimortality has been evidenced, placing it as a more accurate biological indicator of health than level of physical activity [13]. The components of physical fitness related to health are cardiorespiratory fitness, muscular strength, and flexibility [14]. Physical fitness has been measured in Chile by means of the SIMCE test, in which the appraisal was undertaken through structural aspects (muscle and joints functioning) and functional aspects (cardiovascular performance). The results indicated that 2% of students achieved a satisfactory level in the structural aspect, and 28% in the functional aspect. Furthermore, this evaluation reveals that girls presented a worse physical fitness than boys [7].

A widely used parameter to observe appropriate cognitive functioning in school students is the assessment of executive functions, which is a multidimensional construct that encompasses a series of cognitive processes that are necessary to carry out complex tasks, directed to a specific objective [15]. In general terms, executive functioning appears in preschool age and continues to develop throughout childhood and adolescence. This development that takes places during school years can be analyzed through an integrative framework that comprises the main executive functions: working memory, mental flexibility, and self-control [15]. Currently, there are different instruments to measure this indicator, namely the Wisconsin Card Sorting Test (M-WCST), the Behavior Rating Inventory of Executive Function test—Preschool version (BRIEF^®^-P), and one of the most widely used and disseminated instruments in South America, the Battery of Neuropsychological Assessment for Executive Function in Children (ENFEN) [16,17,18,19].

Despite existing studies establishing relationships between physical fitness and executive functions, consensus has not been reached due to the lack of concrete evidence. In the national context, Illesca and Alfaro [20] carried out research in students from a public school in Temuco. This research established that a significant correlation between physical fitness and all cognitive capacities (classification variables, series and letters and numbers from the cognitive skill item), except for perceptual organization, exists [20]. It is essential to keep in mind that childhood is a crucial period for the development of the brain, characterized by extensive maturation of the circuits destined to support brain operations, thus allowing unique opportunities for the optimization of cognitive functions through physical activity [21].

Although studies have been carried out in this area at the international level, studies related to physical fitness markers and their relationship with performance in executive functions tests are scarce in Chile. Moreover, it is critical to identify which executive functions can be profoundly affected by physical fitness, as well as identifying which physical fitness parameters are more closely related to a better executive functioning in school students. Studying both variables, physical fitness and executive functions, is vital in the school context since they are markers that allow us to identify integral health in Chilean students. Currently, national studies indicate the existence of a high percentage of children who have unsatisfactory physical fitness, which could potentially cause a decrease in executive functions in Chilean school students.

Thus, a research question arises: what is the relationship between cardiorespiratory, musculoskeletal, and motor fitness, measured through the ALPHA fitness test, and performance in executive functions, measured through the ENFEN test, in 9- to 13-year-old students from a public school in the south of Chile? To answer this question, the aim was to analyze the relationship between cardiorespiratory, musculoskeletal, motor fitness, and performance in an executive functions test in students from a public school in the south of Chile.

Our hypothesis is that Chilean students analyzed who present a better cardiorespiratory, musculoskeletal, and motor fitness also have a better performance in executive functions tests.

## 2. Materials and Methods

### 2.1. Study Design

Analytical cross-sectional study was conducted following the STROBE guidelines.

### 2.2. Population and Sample

All students from fifth to eight grades of a public school, from the Biobío province, Chile, were invited to participate (*n* = 325). A non-probabilistic sample of students between the ages of 9 to 12 and 11 months (Mean: 11.1; SD: 1.0), of Chilean nationality, and who completed all the measurements after their parents or guardians had consented was included. Initially, 148 students were recruited to participate. Of these, 100 students completed the physical fitness assessments, and 81 students completed the executive function assessments.

### 2.3. Procedure

An alliance between the research team and the district’s education department (DAEM) was formed. The design of the study and the selection of variables were made in conjunction with the schools’ boards. Then, a multiprofessional group was formed by a psychologist, a physical therapist, a physical education teacher, and an occupational therapist, who oversaw the evaluations. After that, the families, teachers, and the school board were informed about the purposes of the study and agreed to take part. The Ethics Committee of Universidad Santo Tomás, Chile, approved the project (protocol code n° 20-19, 27 March 2019), and all procedures were carried out according to the Declaration of Helsinki for research with human subjects.

### 2.4. Outcomes and Instruments of Measurement

#### 2.4.1. Executive Functions

These outcomes were assessed using the Neuropsychological Assessment of Executive Functions Battery for Children (ENFEN) [22]. This instrument assesses global maturity development in children from 6 to 12 years of age, focusing on executive functions controlled by prefrontal brain areas, level of maturity, and cognitive performance in activities related to executive functions in children. Specifically, it analyzes aspects such as vocabulary range, cognitive flexibility, graphomotor and visuomotor coordination, working memory, planning, and sequencing capacity, inhibition capacity, or resistance to interference. It is applied individually, and its duration is approximately 20 min. It is made up of four tests (fluency, trails, rings, and interference) divided into six scales. Lastly, a total score for each task is obtained, and raw scores are transformed into sten scores (M = 5.5, SD = 2).

##### Fluency Task

This test assesses participants’ ability to produce language under time pressure, although it can be an indirect measurement of verbal memory and WM. The child has 1 min to produce as many words as possible from a category: in phonemic fluency, the category was related to words starting with the letter “M” and in semantic fluency “animals”. The resulting direct score is “positive” depending on number of words; the bigger the number of elements provided in both scales, the higher the score.

##### Trail Making Test

This test assesses different aspects of executive functions: flexibility, thinking strategies, inhibition, WM, and executive attention. In the gray trail, the child must draw a line linking numbers from 20 to 1 that appear randomly on a sheet of paper. In the color version, the child must link numbers from 1 to 21, but he/she must switch among yellow and pink colors. The resulting direct score is positive on its direction; the bigger the number of digits accurately linked in the shortest time, the higher the score.

##### Rings

This test assesses executive functions such as planning, sequencing, self-control, and WM. The direction of the direct score is “negative” since the less time it takes for the movements, the higher the score.

##### Inhibition Task

This test is derived from the Stroop test, and it is a relatively pure measure of cognitive inhibition, although attention, flexibility, and resistance to interference are also engaged in this task. The child is presented with a sheet that shows three columns with 13 words each. The words are the names of colors (red, green, yellow, and blue) printed with random color inks (red, green, yellow, and blue), with the color name and color ink never matching. The child must say the color ink of each word. The scoring uses the same criteria as the trail scale since the “right answers” are the number of color words correctly read.

To categorize the results, the mean of students in each of the six scales assessed by the ENFEN test was calculated, and based on that score, data were classified as “below mean” or “above mean”.

#### 2.4.2. Physical Fitness

##### Cardiorespiratory Fitness

This outcome was assessed using the 20 m shuttle run test (20 m SRT), also known as Course Navette. The test consists of running back and forth on a 20 m track marked between two separate lines for as long as possible. The rhythm is set using audio signals. The initial speed is 8.5 km/h and is increased by 0.5 km/h intervals every 1 min. Subjects must step behind the 20 m line when the audio signal or beep is heard. The test finishes when the subject stops because of fatigue or fails to reach the end line concurrent with the beep on two consecutive occasions. Test performance was recorded using the number of 20 m laps (1 lap = 20 m) and total time (s). This test was performed using the protocol given by the Alpha fitness test battery [23].

##### Musculoskeletal Fitness

This outcome was assessed using the standing broad jump (SBJ) and handgrip strength dominant hand.

The SBJ test was used as an indicator of lower limb strength. It consists of jumping the longest possible distance from a standing start, with both feet and swinging both arms. The distance is measured from the takeoff line to the point where the back of the heel nearest to the takeoff line lands on the ground. This test was performed, using the protocol given by the Alpha Fitness test battery [23].

Handgrip strength dominant hand was used as an indicator of upper limb strength (JAMAR (brand) hydraulic dynamometer (Hydraulic Hand Dynamometer^®^ Model PC-5030 J1, Fred Sammons, Inc., Burr Ridge, IL, USA)). Each subject was seated in a standard position on a chair with a straight back. Students were asked to exert pressure on the dynamometer twice with each hand. To control for effects of fatigue, the attempts were performed by alternating the hands with approximately 2 min of rest between each attempt for each hand. The best measurement was recorded for each of the two attempts [23,24].

##### Motor Fitness

This outcome was assessed using the 4 × 10 m shuttle run test. It consisted of running back and forth between two lines 10 m apart taking three sponges alternately as quickly as possible. The total distance run was 40 m. This test was performed using the protocol given by the Alpha fitness test battery [23].

All these tests have been widely used to assess physical fitness in children and adolescents in international and national studies [25]. Test administrators conducted several familiarization trials, and examples of correct and incorrect trials were demonstrated. All tests were conducted in participating schools’ physical education halls and took place during timetabled physical education classes. Tests were administered in small groups of six or fewer participants at a testing station at any one time. Furthermore, to address fatigue or test sequencing as potential sources of measurement error, all participants had a minimum rest period of between three and five minutes between each testing station. The measurements were carried out during a two-week period, and, in some cases, the assessment of a student took two days.

### 2.5. Socio-Educational Data

Additionally, socio-educational data, such as age, sex, grade of each student, and place of residence were recorded.

### 2.6. Statistical Analysis

The data were analyzed by means of the statistical software SPSS 25.0 (IBM SPSS statistics, Chicago, IL, USA). A descriptive analysis including socio-educational (age, sex, grade, and place of residence), physical fitness, and executive functions variables was carried out. The qualitative data were represented through absolute and percentage frequency, while quantitative data were represented through the mean and its corresponding 95% confidence interval (95% CI). The distribution of data was analyzed through a Kolmogorov–Smirnoff test and the equality of variances through a Levene test, which showed a normal distribution and homogeneity of variances, thus leading to a parametric statistical analysis. A one-way ANOVA test was used. In the case of significant differences, a post hoc test (Bonferroni) was used to confirm where the differences occurred between the groups.

A multiple linear regression analysis was used for analyzing the association between physical fitness and executive functions. The effects β_i_ of each single level *i* of these factors were estimated with its corresponding 95% confidence interval. The multivariate model was adjusted for relevant concomitant variables, selected from socio-educational characteristics. Model 0: non-adjusted and Model 1: adjusted for socio-educational variables (sex, age, grade). Significance level: * *p* < 0.05, ** *p* < 0.01, *** *p* < 0.001.

## 3. Results

Table 1 shows socio-educational characteristics of the analyzed students. The ANOVA analysis identified statistically significant differences in cardiorespiratory fitness (Course Navette test (laps; *p* < 0.001)), motor fitness (4 × 10 test; *p* < 0.01), and musculoskeletal fitness (standing broad jump; *p* < 0.01 and dominant handgrip; *p* < 0.001) in students from different grades, which indicates that students from more advanced grades performed better than students from lower grades in cardiorespiratory fitness, motor fitness, and musculoskeletal fitness tests. Accordingly, these same differences were observed in 10–11-year-old and 12–13-year-old students, which indicates that older students achieved better scores in cardiorespiratory fitness, motor fitness, and musculoskeletal fitness tests than younger students.

Table 2 displays the scores obtained in the ALPHA fitness and ENFEN tests. It is evidenced that in the ENFEN test results, 34.58% of students showed a poor level of phonological fluency, and 55.69% of students achieved below mean scores in the rings test. Moreover, no participant reached a high or very high level in the rings test. For the rest of the tests, most students were located on an average level.

Table 3 portrays the physical fitness of students, according to scores achieved in the ENFEN test. Statistically significant differences were identified in cardiorespiratory fitness (Course Navette test), motor fitness (4 × 10 test), and musculoskeletal fitness (standing broad jump and dominant handgrip tests) in students who reached below mean and above mean scores in the gray trail and colored trail components of the ENFEN test. The results reveal that students who achieved above mean scores in the gray trail test performed better in the Course Navette tests (stage; *p* < 0.05; time; *p* < 0.05), standing broad jump tests (centimeters; *p* < 0.05), and 4 × 10 tests (seconds; *p* < 0.05). Furthermore, students with above mean scores in the colored trail tests performed better in the dominant handgrip tests (kg; *p* < 0.05).

In line with the results shown in Table 3, in a non-adjusted model (model 0), students with below mean scores in the gray trail test endured fewer stages (β = −7.23; *p* < 0.05) and lasted a shorter time (β = −52.5; *p* < 0.05) in the 20 m SRT, took longer in the 4 × 10 test (β = 0.78; *p* < 0.05), and jumped fewer centimeters in the SBJ test (β = −12.7; *p* < 0.05), as opposed to the students with above mean scores in the gray trail test. Moreover, it was established that students with below mean scores in the colored trail test applied less strength in the dominant handgrip test (β = −4.06; *p* < 0.05) when compared to students with above mean scores in the gray trail test. These results were consistent with a model adjusted for confounding socio-educational variables (sex, age, grade) (Table 4).

## 4. Discussion

### 4.1. Main Findings of the Study

A relationship between performance in the Course Navette test, 4 × 10 test, standing broad jump, and dominant handgrip test and scores in the gray trail and colored trail, both components of the ENFEN test, was observed. This result suggests that students who exhibit better cardiorespiratory, motor, and musculoskeletal fitness measured by the ALPHA fitness test, present a better performance of executive functions such as flexibility, thinking strategies, inhibition, working memory, and executive attention, measured by the ENFEN test.

### 4.2. Does Similar Evidence to This Study Exist?

Previous studies have evaluated the health-related quality of life in the child population [26,27] and have related it to physical condition [10]. In addition, recent studies have referred to the relationship between physical fitness markers and performance in executive functions tests. In the local context, a study carried out by Illesca and Alfaro [20] studied Chilean children and evidenced a dependency relationship between physical fitness and letter–number sequencing capacity, measured by means of the six-minute walk test and the Psych pedagogical Evaluation Battery [20]. Although other evaluation methods were used, the results of the study are similar to ours in the ENFEN trails tests, which use numerical sequences. On the other hand, in the international context, Moradi et al. [28] researched the relationship between these variables in 10- to 12-year-old Iranian children, using different tests to measure physical fitness, and the Simon task to measure inhibitory control, which unveiled a positive association between agility and inhibitory control [28], coinciding with our results. Likewise, Nieto-López et al. [29], who studied Spanish children, established a direct relationship between inhibitory control, assessed through the NIH Toolbox, and cardiorespiratory fitness, assessed through the PREFIT battery [29]. In addition to this, previous studies have indicated that physical fitness, more specifically muscle aptitude, brings health benefits for children, including decreases in adiposity and cardiometabolic risk, which are associated with a better cognitive control, mainly working memory [30].

At the international level, a study of 70 Chinese children showed how inhibitory control and working memory improved in those who improved their physical condition after 12 weeks of training (agility and speed) [31]. On the other hand, and after a 12-week intervention in 90 schoolchildren, based on the development of complex motor skills in handball and athletics, executive functions improved significantly [32].

In a study of 10–12-year-olds, they observed higher levels of physical activity, better sprint performance, greater lower body muscle power, and greater upper body muscle strength, and these were associated with better working memory, cognitive flexibility, inhibition, planning, and/or attention [33].

On the other hand, the effect of a multimodal classroom program of physical activity and executive function in New Jersey was evaluated, increasing executive conditions after the intervention [34]. In one study, the association between cardiorespiratory fitness and executive function performance in 132 Brazilian adolescents aged 11 to 16 years were investigated, and were related to better planning skills, problem solving, and cognitive flexibility as well as body mass index with inhibitory control [35].

Muscle strength, speed agility, and cardiorespiratory fitness were associated with executive function in 100 overweight and obese children. Cognitive flexibility and inhibition appeared to be most strongly associated with all fitness components, speed, and agility, while planning ability with muscle strength, speed agility, and cardiorespiratory fitness [36]. In a cross-sectional study of 3378 9–10-year-olds from nine primary schools in Stavanger, Norway, they showed the relationships between aerobic fitness and mathematical performance and how it was mediated by executive functions [37].

However, and despite previous studies, our study is a pioneer in the study of children in Chile and directly assesses the relationship of physical fitness, unlike the studies on executive function skills.

Additionally, our results, as well as those of Moradi et al. [28] and Nieto-López et al. [29] pinpointed a significant relationship between physical fitness and inhibitory control. One of the hypotheses that could explain this is that cardiorespiratory fitness promotes angiogenesis in the motor cortex and increases blood flood, thus improving brain vascularization, which can positively influence cognition [38].

### 4.3. Limitations of the Study

Executive functions are complex mental activities whose evaluation in children and adolescents can be carried out through multiple instruments. Therefore, poor performance in executive functions tests can be caused by several risk factors that can occur simultaneously. In this sense, the design of the study (analytical, cross-sectional study) allowed for analysis of the association between performance in executive functions, through the ENFEN test, and a battery of tests to evaluate physical fitness related to health, the ALPHA fitness test, but it did not determine the causes of poor performance. Moreover, social variables such as race, ethnicity, or place of residence; current health condition; genetic inheritance; and personal, family, or contextual characteristics that could be related to poor performance in executive functions were not analyzed.

Another limitation is that the sample is not representative, hence, the results obtained cannot be generalized to the entire population from a public school in the south of Chile. However, these results can shed light on the relationship between physical condition and executive functions in Chilean schoolchildren and contribute to the scarce existing evidence at the national and Latin American level.

### 4.4. Contribution to the Discipline, Practical Implications, and Future Lines of Research

#### 4.4.1. Theoretical Contributions

This study contributes with data about the relationship between performance in executive functions tests and physical fitness markers in school students. According to the evidence gathered, this study is a pioneer in Chile and Latin America on this topic. To the best of our knowledge, our study is the first study in Latin America to analyze the association between performance in executive functions, measured through the ENFEN test, and physical fitness markers, measured through the ALPHA fitness test. Thus, this study represents the most recent evidence showing that students from a public school in the south of Chile who obtain a lower score on the gray trail test and colored trail test, both ENFEN scales, also present a poor performance in cardiorespiratory, musculoskeletal, and motor fitness tests.

#### 4.4.2. Practical Implications

From a practical point of view, the results might be of use to professionals and school boards, to reflect on the possible negative impact that poor physical fitness may have on the development of executive functions. Furthermore, this study is relevant since it raises awareness about the importance of assessing and monitoring physical fitness and executive functions in school students systematically. Finally, the findings pose the need for the development of future programs oriented towards the early identification of risk populations with poor physical fitness, low levels of physical activity, and unhealthy habits or routines since these might be relevant in the development of executive functions, cognitive functions, and academic performance.

#### 4.4.3. Future Lines of Research

Future studies could focus on studying whether the relationship between executive functions and physical fitness performance is bidirectional or unidirectional. In addition, it could be analyzed if the relationship is outcome-dependent or if this relationship is maintained in older students.

## 5. Conclusions

This study strengthens existing evidence pointing to a relationship between cardiorespiratory, motor, and musculoskeletal fitness, measured through the ALPHA fitness test, and performance in executive functions, measured through the ENFEN test, in 9- to 13-year-old Chilean public school students.

Concretely, these results propose that students who achieved more stages and a longer time on the Course Navette test, more centimeters on the standing broad jump test, and less time on the 4 × 10 SRT, obtained higher scores in the gray trail test. Furthermore, it was evidenced that students with a stronger handgrip scored higher on the colored trail test.

Researchers are encouraged to delve deep into the relationship that the studied variables can have and the direction of said relationship. Maintaining adequate levels of physical fitness and promoting healthy lifestyle habits in children is related to better executive function skills, and therefore, cognitive skills that will help children to achieve a goal, achievement, or target at school age. Therefore, it is necessary to implement appropriate policies for the good maintenance of children’s physical condition, given its relationship with the planning of activities and tasks, decision making, motor or cognitive behavioral inhibition, organization, cognitive flexibility, and anticipation, among other abilities given by the good development of executive functions. Bearing this knowledge in mind, it implies that physical fitness improves later academic performance at school age, if we consider the function of executive skills at school age; so improving physical fitness could be a variable to be considered in those students with low academic performance. Hence, it is important to identify if these associations depend on the instruments, scales, or specific tests used since this study could only identify a link between the trail making tests and physical fitness. Moreover, this study is expected to work as a starting point for future experimental studies that could possibly explain this connection.

## Figures and Tables

**Table 1 behavsci-13-00191-t001:** Socio-educational characteristics of the sample.

		Cardiorespiratory Fitness		Motor Fitness	Musculoskeletal Fitness	
		20 m SRT (Laps)	20 m SRT (Time in Seconds)	Test 4 × 10 (s)	SBJ (cm)	Handgrip Strength Dominant Hand (kg)
Factors (*Levels*)	%	Mean [CI 95%]	Mean [CI 95%]	Mean [CI 95%]	Mean [CI 95%]	Mean [CI 95%]
Sex						
*Male* *Female*	44 56	23.98 [18.41; 29.54] 18.71 [16.09; 21.34]	185.12 [145.01; 225.23] 151.93 [131.94; 171.93]	13.75 [13.31; 14.21] 14.10 [13.74; 14.48]	115.19 [106.82; 123.56] 107.46 [102.78; 112.15]	20.11 [18.48; 21.75] 21.34 [20.09; 22.60]
Grade						
*Fifth* *Sixth* *Seventh*	44 28 28	16.44 [13.01; 19.87] ^a^ 19.85 [14.72; 24.98] ^a^ 29.41 [23.07; 35.75] ^b^	131.90 [106.43; 157.37] ^a^ 160.61 [122.66; 198.55] ^a^ 226.90 [181.76; 272.04] ^b^	14.54 [14.08; 15.00] ^b^ 13.74 [13.28; 14.20] ^a^ 13.23 [12.80; 13.67] ^a^	104.12 [96.58; 111.66] ^a^ 112.14 [105.44; 118.85] ^b^ 120.18 [112.03; 128.32] ^b^	18.34 [17.11; 19.57] ^a^ 20.83 [19.04; 22.62] ^b^ 24.72 [23.09; 26.35] ^c^
Place of residency						
*Rural* *Urban*	6 94	13.08 [ 5.56; 20.61] 21.53 [18.56; 24.52]	109.17 [49.17; 169.16] 170.20 [148.49; 191.91]	13.61 [12.22; 15.01] 13.97 [13.68; 14.27]	110 [96.40; 123.60] 110.91 [106.17; 115.67]	21.25 [14.63; 27.87] 20.77 [19.76; 21.79]
Age						
*10–11 years old* *12–13 years old*	30 70	15.65 [11.58; 19.72] ^a^ 23.33 [19.74; 26.93] ^b^	125.32 [95.52; 155.12] ^a^ 184.20 [158.14; 210.27] ^b^	14.61 [14.02; 15.21] ^b^ 13.66 [13.37; 13.97] ^a^	108.26 [101.54; 115.0] 111.97 [106.15; 117.8]	17.87 [16.42; 19.32] ^a^ 22.06 [20.88; 23.24] ^b^

Caption: The statistical analysis was carried out through a one-factor ANOVA. In the same row abc marks with different symbols indicate statistically significant differences between groups. (One-factor ANOVA and Bonferroni post hoc test).

**Table 2 behavsci-13-00191-t002:** Scores obtained on the physical fitness ALPHA test and the ENFEN test.

Variables	Measurement Instruments	
Physical fitness (ALPHA Fitness) (*n* = 100)		M (SD)
*Cardiorespiratory fitness*	* Test Course Navette (laps)*	21.03 (14.34)
	* Test Course Navette (time in seconds)*	166.53 (104.55)
*Motor fitness*	* 4 × 10 SRT (m)*	13.95 (1.43)
*Musculoskeletal fitness*	* Standing Broad Jump test (cm)*	110.86 (22.68)
	* Handgrip (dominant hand) (kg)*	20.80 (5.02)
Executive functions (ENFEN) (*n* = 81)		
Verbal memory and WM	* Phonological fluency*	8.87 (3.14)
	* Semantic fluency*	14.39 (3.25)
Flexibility, thinking strategies, inhibition, WM, and EX	* Gray trail test*	27.1 (7.68)
	* Colored trail test*	15.02 (5.39)
Planning, sequencing, self-control, and WM	* Rings test*	185.10 (44.35)
Attention, flexibility, and resistance to interference	*Interference test*	74.56 (21.74)
	ENFEN classification	
	Phonological fluency	Frequency (%)
	*Very low*	28 (34.58%)
	*Low*	21 (25.93%)
	*Medium–low*	13 (16.05%)
	*Medium*	16 (19.75%)
	*Medium–high*	2 (2.47%)
	*Very high*	1 (1.23%)
	Semantic fluency	
	*Very low*	14 (17.28%)
	*Low*	14 (17.28%)
	*Medium–low*	20 (24.69%)
	*Medium*	27 (34.34%)
	*Medium–high*	5 (6.17%)
	*High*	1 (1.23%)
	Gray trail test	
	*Very low*	4 (5%)
	*Low*	8 (10%)
	*Medium–low*	15 (18.75%)
	*Medium*	27 (33.75%)
	*Medium–high*	14 (17.50%)
	*High*	9 (11.25%)
	*Very high*	3 (3.75%)
	Colored trail test	
	*Very low*	16 (20%)
	*Low*	13 (16.25%)
	*Medium–low*	11 (13.75%)
	*Medium*	32 (40%)
	*Medium–high*	5 (6.25%)
	*High*	1 (1.25%)
	*Very high*	2 (2.5%)
	Rings test	
	*Very low*	12 (15.19%)
	*low*	16 (20.25%)
	*Medium–low*	16 (20.25%)
	*Medium*	30 (37.98%)
	*Medium–high*	5 (6.33%)
	Interference test	
	*Very low*	20 (25.32%)
	*low*	8 (10.13%)
	*Medium–low*	9 (11.39%)
	*Medium*	29 (36.71%)
	*Medium–high*	8 (10.13%)
	*High*	3 (3.80%)
	*Very high*	2 (2.54%)

WM: working memory, EX: executive attention.

**Table 3 behavsci-13-00191-t003:** Physical fitness according to ENFEN test score.

		Cardiorespiratory Fitness		Motor Fitness	Musculoskeletal Fitness	
		20 m SRT (Laps)	20 m SRT (Time in Seconds)	4 × 10 Test (s)	SBJ (cm)	Handgrip Strength Dominant Hand (kg)
Variables	*n* (%)	M [CI 95%]	M [CI 95%]	M [CI 95%]	M [CI 95%]	M [CI 95%]
Phonological fluency						
*Below mean (1–6)*	71 (97.26%)	18.92 [16.00;21.83]	151.86 [130.06;173.68]	14.14 [13.80;14.47]	106.55 [101.25;111.86]	19.84 [18.66;21.02]
*Above mean (7–10)*	2 (2.74%)	16.25 [13.07;19.42]	135.0 [135;135]	14.07 [6.76;21.38]	98.0 [−16.36;212.36]	16.2 [−21.91;54.31]
Semantic fluency						
*Below mean (1–6)*	68 (93.15%)	18.91 [15.89;21.93]	151.61 [129.01;174.21]	14.17 [13.83;14.51]	106.00 [100.50;111.51]	19.61 [18.41;20.81]
*Above mean (7–10)*	5 (6.85%)	18.0 [9.61;26.38]	148.6 [83.71;213.49]	13.64 [11.68;15.60]	110.6 [93.05;128.15]	21.58 [15.16;27.99]
Gray trail test						
*Below mean (1–6)*	50 (69.44%)	16.79 [13.63;19.94] *	136.52 [112.85;160.19] *	14.38 [13.95;14.81] *	102.56 [97.09;108.03] *	19.40 [19.87;20.94]
*Above mean (7–10)*	22 (30.56%)	24.02 [18.17;29.87]	189.02 [145.14;232.91]	13.59 [13.15;14.03]	115.29 [103.52;127.06]	20.55 [18.78;22.33]
Colored trail test						
*Below mean (1–6)*	65 (90.27%)	18.75 [15.74;21.76]	150.37 [127.85;172.91]	14.14 [13.78;14.50]	106.45 [100.86;112.04]	19.36 [18.12;20.59] *
*Above mean (7–10)*	7 (9.73%)	21.28 [9.29;33.28]	172.85 [83.74;261.97]	14.13 [13.07;15.19]	106.42 [86.24;126.61]	23.47 [20.25;26.68]
Rings test						
*Below mean (1–6)*	66 (92.95%)	18.75 [15.77;21.74]	150.61 [128.34;172.89]	14.18 [13.83;14.54]	105.32 [99.85;110.80]	19.59 [18.38;20.80]
*Above mean (7–10)*	5 (7.05%)	22.7 [4.59;40.80]	180.8 [42.30;319.30]	13.78 [12.44;15.13]	117.8 [88.79;146.81]	22.92 [16.06;29.77]
Interference test						
*Below mean (1–6)*	60 (84.5%)	18.85 [15.52;22.18]	151.44 [126.65;176.23]	14.18 [13.80;14.56]	105.22 [99.10;111.34]	20.20 [18.88;21.52]
*Above mean (7–10)*	11 (15.5%)	20.04 [14.49;25.59]	159.81 [115.39;204.24]	14.05 [13.27;14.83]	111.54 [102.91;120.18]	17.80 [15.08;20.53]

* *p* < 0.05.

**Table 4 behavsci-13-00191-t004:** Association between executive function and performance in physical fitness.

Factor/Levels	Cardiorespiratory Fitness		Motor Fitness	Musculoskeletal Fitness	
	20 m SRT (Laps)	20 m SRT (Time in Seconds)	4 × 10 Test (s)	SBJ (cm)	Handgrip Strength Dominant Hand (kg)
Model 0					
	β_i_ [CI 95%]	β_i_ [CI 95%]	β_i_ [CI 95%]	β_i_ [CI 95%]	β_i_ [CI 95%]
Phonological fluency					
*Below mean (1–6)* *Above mean (7–10)*	2.67 [−0.98; 33.48] Reference	16.87 [−113.9; 147.6] Reference	0.066 [−1.96; 2.09] Reference	8.55 [−23.3; 40.4] Reference	3.68 [−3.34; 10.7] Reference
Semantic fluency					
*Below mean (1–6)*	0.91 [−10.38; 12.21]	3.01 [−81.5; 87.5]	0.53 [−0.77; 1.83]	−4.59 [−25.2; 16.0]	−1.92 [−6.48; 2.63]
*Above mean (7–10)*	Reference	Reference	Reference	Reference	Reference
Gray trail test					
*Below mean (1–6)* *Above mean (7–10)*	−7.23 [−13.24; −1.22] * Reference	−52.5 [−97.5; −7.4] * Reference	0.78 [0.08; 1.49] * Reference	−12.7 [−23.8; −1.6] * Reference	−1.12 [−3.65; 1.41] Reference
Colored trail test					
*Below mean (1–6)*	−2.53 [−12.23; 7.17]	−22.5 [−95.0; 50.0]	0.014 [−1.11; 1.14]	−0.02 [−17.8; 17.8]	−4.06 [−7.9; −0.23] *
*Above mean (7–10)*	Reference	Reference	Reference	Reference	Reference
Rings test					
*Below mean (1–6)*	−3.94 [−15.32; 7.43]	−30.18 [−115.3; 54.9]	0.40 [−0.91; 1.72]	−12.4 [−33.1; 8.18]	−3.28 [−7.8; 1.26]
*Above mean (7–10)*	Reference	Reference	Reference	Reference	Reference
Interference test					
*Below mean (1–6)*	−1.20 [−9.26; 6.87]	−8.37 [−68.7; 51.9]	0.13 [−0.80; 1.06]	−6.3 [−21.0; 8.36]	2.44 [−0.77; 5.65]
*Above mean (7–10)*	Reference	Reference	Reference	Reference	Reference
**Model 1 (adjusted for sociodemographic variables)**					
	β_i_ [CI 95%]	β_i_ [CI 95%]	β_i_ [CI 95%]	β_i_ [CI 95%]	β_i_ [CI 95%]
Phonological fluency					
*Below mean (1–6)* *Above mean (7–10)*	5.0 [−12.82; 22.8] Reference	33.65 [−99.2; 166.5] Reference	0.016 [−2.0; 2.03] Reference	13.16 [−18.5; 44.8] Reference	4.84 [−1.86; 11.5] Reference
Semantic fluency					
*Below mean (1–6)*	1.93 [−9.28; 13.24]	11.36 [−72.9; 95.7]	0.37 [−0.89; 1.65]	−4.6 [−24.7; 15.5]	−1.21 [−5.51; 3.09]
*Above mean (7–10)*	Reference	Reference	Reference	Reference	Reference
Gray trail test					
*Below mean (1–6)* *Above mean (7–10)*	−7.42 [−13.57; −1.26] * Reference	−54.46 [−100.3; −8.55] * Reference	0.95 [0.26; 1.64] * Reference	−13.45 [−24.4; −2.5] * Reference	−1.45 [−3.87; 0.97] Reference
Colored trail test					
*Below mean (1–6)*	−0.89 [−120.8; 9.01]	−9.18 [−82.9; 64.6]	−0.27 [−1.39; 0.84]	1.53 [−16.1; 19.2]	−2.87 [−6.6; 0.85] *
*Above mean (7–10)*	Reference	Reference	Reference	Reference	Reference
Rings test					
*Below mean (1–6)*	−1.77 [−13.5; 9.95]	−13.23 [−100.6; 74.1]	0.24 [−1.07; 1.55]	−13.02 [−33.3; 7.28]	−2.19 [−6.6; 2.23]
*Above mean (7–10)*	Reference	Reference	Reference	Reference	Reference
Interference test					
*Below mean (1–6)*	−1.03 [−9.68; 7.61]	−8.26 [−72.7; 56.2]	0.28 [−0.68; 1.25]	−2.46 [−17.6; 12.7]	2.55 [−0.68; 5.78]
*Above mean (7–10)*	Reference	Reference	Reference	Reference	Reference

* *p* < 0.05.

## Data Availability

Data will be made available upon request.

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
