# Peer review of "Students from a Public School in the South of Chile with Better Physical Fitness Markers Have Higher Performance in Executive Functions Tests—Cross-Sectional Study"

_behavsci, 2023, doi:10.3390/bs13020191_

Round 1

Reviewer 1 Report (Previous Reviewer 1)

The concept of finding the association between physical fitness level and executive functions still lack the scientific justification. The authors should consider a specific spectrum of level of physical function and cognitive functioning. Every motor task require cognitive participation. The aim of the study and what authors wish to prove and how the community is going to get the benefit is still unclear.  

Author Response

Reviewer 1

The concept of finding the association between physical fitness level and executive functions still lack the scientific justification. The authors should consider a specific spectrum of level of physical function and cognitive functioning. Every motor task require cognitive participation. The aim of the study and what authors wish to prove and how the community is going to get the benefit is still unclear.

Answer 1: Thank you very much for comments, the entire manuscript has been reviewed, and several changes have been made; title, aim, tables, results, and discussion have been modified. Now, that association is more specific.

Besides, benefits and practical contributions have been clarified.

Reviewer 2 Report (Previous Reviewer 2)

Accepted

Author Response

Reviewer 2

Accepted

Answer: thank you very much.

Reviewer 3 Report (New Reviewer)

The work is interesting and deals with an important problem of children's development, and the authors should supplement the work with the concept of Health-Related Quality of Life, which includes both physical and intellectual development, in the introduction. In the discussion, they should refer to the results obtained also in measuring the Health-Related Quality of Life people children, among others, in the works of Chmielik LP, Mielnik-Niedzielska G, Kasprzyk A, Stankiewicz T, Niedzielski A. Physical and Psychosocial Concept Domains Related to Health-Related Quality of Life (HRQL) in 50 Girls and 52 Boys Between 5 and 18 Years Old in Poland Using the Parent-Reported 50-Item Child Health Questionnaire (CHQ-PF50). Med Sci Monit. 2022 Jun 9;28:e936801. doi: 10.12659/MSM.936801. and Ruperto N, Ravelli A, Pistorio A, Cross-cultural adaptation and psychometric evaluation of the Hildehood Health Assessment Questionnaire (CHAQ) and the Hilde Health Questionnaire (CHQ) in 32 countries. Review of the general methodology: Clin Exp Rheumatol, 2001; 19(Suppl 23); 1-9

Author Response

Comment 1: The work is interesting and deals with an important problem of children's development, and the authors should supplement the work with the concept of Health-Related Quality of Life, which includes both physical and intellectual development, in the introduction.

Answer 1: thanks for your comments. The concept of Health-Related Quality of Life has been included in introduction and discussion.

Comment 2:  In the discussion, they should refer to the results obtained also in measuring the Health-Related Quality of Life people children, among others, in the works of:

  • Chmielik LP, Mielnik-Niedzielska G, Kasprzyk A, Stankiewicz T, Niedzielski A. Physical and Psychosocial Concept Domains Related to Health-Related Quality of Life (HRQL) in 50 Girls and 52 Boys Between 5 and 18 Years Old in Poland Using the Parent-Reported 50-Item Child Health Questionnaire (CHQ-PF50). Med Sci Monit. 2022 Jun 9;28:e936801. doi: 10.12659/MSM.936801.
  • Ruperto N, Ravelli A, Pistorio A, Cross-cultural adaptation and psychometric evaluation of the childhood Health Assessment Questionnaire (CHAQ) and the Hilde Health Questionnaire (CHQ) in 32 countries. Review of the general methodology: Clin Exp Rheumatol, 2001; 19(Suppl 23); 1-9

Answer 2: thank for you comment, three references have been added in the discussion section (the two references suggested by the reviewer are included).

Round 2

Reviewer 3 Report (New Reviewer)

Good work

I accept in present form

This manuscript is a resubmission of an earlier submission. The following is a list of the peer review reports and author responses from that submission.

Round 1

Reviewer 1 Report

The concept in not innovative.

Objective outcome measures need standardization.

The objective of the study is not clear.

The significance for the study is unclear.

What was the need to find the association between physical fitness and executive functions?

Sample size is very less.

Reviewer 2 Report

Relationship between physical fitness and executive functions in students from a public school in the south of Chile

First of all, the reviewer would like to thank the authors for their work and efforts in trying to improve sports science knowledge.

General comments to the authors

The article is investigating the relationship between physical fitness and executive functions in students from a public school in the south of Chile. Overall, the study is well-designed and well-written, with a great introduction proposing the usefulness of the topic and a clear outline of the research question. I suggest that the author modify/include some suggestions in order to improve the manuscript prior to being published:

Abstract

This section is well-designed and well-written.

Introduction section

This section is well-designed and well-written.

Methods section

The number of ethical files should be added.

In Tables 1, 2, 3and 4; 20-m SRT (time) means what? Seconds or minutes?

Discussion section

Overall the discussion is well-written and incorporates relevant literature.

Tables and Figures

These sections are well-designed and well-written.

Limitations of the study

Is maturation a key factor in the study? Please add maturation status as a limitation for your participants.